# Progress on the Link between Nutrient Availability and Toxin Production by *Ostreopsis* cf. *ovata*: Field and Laboratory Experiments

**DOI:** 10.3390/toxins15030188

**Published:** 2023-03-02

**Authors:** Noemí Inmaculada Medina-Pérez, Elena Cerdán-García, Francesc Rubió, Laia Viure, Marta Estrada, Encarnación Moyano, Elisa Berdalet

**Affiliations:** 1Department of Chemical Engineering and Analytical Chemistry, University of Barcelona, Av. Diagonal 645, E-08028 Barcelona, Spain; 2Department of Marine Biology and Oceanography, Institute of Marine Sciences (ICM-CSIC), Pg. Marítim de la Barceloneta, 37-49, E-08003 Barcelona, Spain; 3Aix Marseille Université, Université de Toulon, CNRS, IRD, MIO UM 110, 13288 Marseille, France; 4Water Research Institute (IdRA), University of Barcelona, Montalegre 6, E-08001 Barcelona, Spain

**Keywords:** *Ostreopsis* cf. *ovata*, isobaric palytoxin, ovatoxins, nutrients

## Abstract

This study aimed to improve the understanding of the nutrient modulation of *Ostreopsis* cf. *ovata* toxin content. During the 2018 natural bloom in the NW Mediterranean, the total toxin content (up to ca. 57.6 ± 7.0 pg toxin cell^−1^) varied markedly. The highest values often coincided with elevated *O.* cf. *ovata* cell abundance and with low inorganic nutrient concentrations. The first culture experiment with a strain isolated from that bloom showed that cell toxin content was higher in the stationary than in the exponential phase of the cultures; phosphate- and nitrate-deficient cells exhibited similar cell toxin variability patterns. The second experiment with different conditions of nitrogen concentration and source (nitrate, urea, ammonium, and fertilizer) presented the highest cellular toxin content in the high-nitrogen cultures; among these, urea induced a significantly lower cellular toxin content than the other nutrient sources. Under both high- and low-nitrogen concentrations, cell toxin content was also higher in the stationary than in the exponential phase. The toxin profile of the field and cultured cells included ovatoxin (OVTX) analogues -a to -g and isobaric PLTX (isoPLTX). OVTX-a and -b were dominant while OVTX-f, -g, and isoPLTX contributed less than 1-2%. Overall, the data suggest that although nutrients determine the intensity of the *O.* cf. *ovata* bloom, the relationship of major nutrient concentrations, sources and stoichiometry with cellular toxin production is not straightforward.

## 1. Introduction

The biogeographic distribution of some harmful species of the benthic dinoflagellate genus *Ostreopsis* seems to be expanding from tropical to temperate waters, and recurrent blooms have been reported in certain beaches of the Mediterranean Sea since the end of the 20th century [1]. Impacts on marine fauna associated with *Ostreopsis* spp. blooms have been documented [2,3], and ecotoxicology studies are underway to elucidate the essential noxious mechanisms involved [4,5]. Palytoxin (PLTX) analogues (namely, putative or isobaric PLTX (isoPLTX) and ovatoxins (OVTXs)) synthesized by *Ostreopsis* species have been associated with sporadic but dramatic seafood-borne poisonings in the tropics [6,7,8] and, most commonly, with cutaneous and respiratory irritations in users of temperate beaches [9,10,11]. While a direct link between the toxic compounds and human health problems has not yet been clearly demonstrated [8], the recurrent *Ostreopsis* spp. blooms seem to be the most plausible cause of the respiratory diseases reported in certain beaches [12,13,14]. This fact, along with the detection of OVTXs and isoPLTXs in some marine fauna of the Mediterranean Sea [15,16,17,18], has encouraged active monitoring to prevent impacts on the health of beach users as well as intensive research to elucidate the factors influencing the blooms and toxicity of *Ostreopsis* [19,20].

On the Mediterranean coasts, *Ostreopsis* cf. *ovata* is the most abundant and bloom-prone species of the genus. Recurrent seasonal blooms of cells attached to biotic and abiotic benthic surfaces by a self-produced mucopolysaccharide substance occur during the summer (in the NW Mediterranean) and fall (in the Adriatic) seasons [21,22,23,24,25] in shallow, well-illuminated, and relatively sheltered waters. As part of the life cycle, in combination with a certain degree of water turbulence, cell aggregates are released from the benthos and float [26]. As in most natural events, different environmental factors are involved in the dynamics of *Ostreopsis* blooms, but a combination of temperature, water motion [27,28], and nutrients [29] appears to be the key players. Some studies also suggest that *O.* cf. *ovata* can grow in habitats subjected to anthropic pressures, including eutrophic waters [25,30,31]. Although a direct link with high nutrient concentrations has not yet been clearly demonstrated [3,25,28], in the bloom-affected areas, the presence of dense macroalgal carpets and high epiphytic microalgal cell densities are indicators of such nutrient availability.

It is widely recognized that the concentration of essential nutrients (i.e., nitrogen (N), phosphorus (P), silicate (Si)) and their ratios play a major role in the occurrence of high-biomass harmful algal blooms (HABs) [32,33,34]. However, how nutrients modulate toxin production is a complex, not very well understood process [35,36,37,38,39]. In a thorough revision of the available literature data, Van de Waal et al. (2014) found that N or P limitation promoted an increased cell quota of carbon-rich toxins (e.g., brevetoxin, karlotoxin) [39]. In that study, data on PLTX analogues, characterized by high molecular weight and carbon content, were not available. However, a more recent conceptual model by Pinna et al. (2015) suggested that the toxins produced by *O.* cf. *ovata* could increase under N- and P-limitation [40]. This model was based on the strong influence of the internal carbon (C) to N or P ratios (C:N or C:P, respectively) on toxin biosynthesis. Indeed, several studies with batch cultures of *O.* cf. *ovata* in the laboratory showed that cellular toxin content increased steadily from the exponential to the stationary phase, when nutrients become depleted [37,41,42,43]. Moreover, Vidyarathna and Granéli (2013) observed significantly higher toxicity (measured as hemolytic activity) in cultures grown in N-deficient conditions (N:P = 1.6) [44]. However, Vanucci et al. (2012) found that *O.* cf. *ovata* cells in the stationary phase of cultures grown under N and P-deficient conditions had, respectively, 53% and 40% lower toxin content than those in the same phase of nutrient-replete cultures [37]. These apparently contradictory results may be due to factors such as the different *O.* cf. *ovata* strains studied and the various experimental designs and toxin analytical methods used, as noted by the aforementioned studies. In this context, Pezzolesi et al. (2016) pointed out that the variability in toxicity and growth dynamics observed in blooms from different coastal areas could be due to the different regimes of nutrient supply [43]. This overall situation indicates the need to further investigate how nutrient conditions relate to cellular toxin content during natural *Ostreopsis* blooms and to explore the responses of different *Ostreopsis* strains.

With this aim, the present study examined in detail the variability and profile of the toxin content of *Ostreopsis* cf. *ovata* in the field and in cultures grown under different nutrient conditions. First, a natural bloom in the NW Mediterranean was monitored in 2018. Determinations of nutrient concentrations in the seawater and of the toxin content of the cells were conducted. In addition, a strain isolated from this bloom was used to perform experiments to investigate how nutrient conditions (including different concentrations of P and inorganic and organic forms of N) might affect its growth and toxin content. In the first series of experiments, the nitrate and phosphate conditions were the same as those used by Vanucci et al. (2012) with an *O.* cf. *ovata* strain isolated from the Adriatic Sea [37], with the aim to facilitate the comparison between the two studies. In the second series of experiments, the NW Mediterranean strain was grown on a variety of N sources, namely, nitrate, ammonium, urea, and a commercialized fertilizer rich in amino acids. This fourth treatment was included because high organic N concentrations, likely originating from agricultural activity and wastewater, were found in the seawater at the study site. The objectives of the present study are twofold: (1) to increase the knowledge of the toxin profiles in different *O.* cf. *ovata* strains from the Mediterranean Sea [45] and the role of nutrients in toxin production and (2) to estimate the toxin content in an area affected by recurrent *Ostreopsis* blooms and the subsequent potential risk of toxin transfer through the food web, ultimately affecting humans.

## 2. Results

### 2.1. Ostreopsis cf. ovata Bloom 2018

Taking as reference the epiphytic concentrations of *Ostreopsis* cf. *ovata* (Figure 1a), the bloom started at the end of June 2018 and developed exponentially through July with a peak (up to ca. 2 × 10^6^ cells gFW^−1^) between July 24th and August 1st. Thereafter, the bloom entered a stationary phase until mid-September, with oscillations in cell numbers, reaching some high values in the 1 × 10^5^–8 × 10^5^ cells gFW^−1^ range. The bloom decreased progressively in autumn. The highest *O.* cf. *ovata* concentrations (around 10^5^ cells L^−1^) in the plankton samples (not shown) coincided with the highest abundances of the epiphytic cells. In the water, high nitrate (22.1–52.3 µM) and ammonium (2.8 µM) concentrations (Figure 1b and Figure 1c, respectively) were found at the initial stage of the bloom and gradually decreased thereafter, with organic N levels remaining between 8.4 and 12.3 µM throughout the bloom (Figure 1b). Concentrations of phosphate (Figure 1c) and organic P (not shown) did not exceed 0.11 µM and 0.23 µM, respectively, throughout the whole sampling period. The resulting N:P ratios, considering the inorganic and organic forms, were mostly above 16.8, i.e., higher than the Redfield ratio. Total organic carbon (TOC) fluctuated throughout the bloom (between 78.0 and 123.6 µM, Figure 1d), resulting in C:N ratios always below 11.

The total cell toxin content of the epiphytic *O.* cf. *ovata* population was below detection at the start and the end of the samplings and fluctuated between 12.2 and 64.7 pg_tox_ cell^−1^ throughout the proliferation (Figure 1a) with several concentration peaks during the exponential, early, and late stationary phases of the bloom.

The total toxin content in the macroalgal surface covered by the epiphytic population (expressed as µg total toxin gFW^−1^) increased in parallel (r = 0.78) to the *O.* cf. *ovata* cell concentration (expressed as cells gFW^−1^).

The statistical relationships among toxin content, cell abundance, and environmental variables were explored by means of correlation coefficients (Appendix A). Although the results must be interpreted with caution, given the reduced number of data points, some patterns stand out. The toxin content per cell was significantly negatively correlated with inorganic nutrient concentrations, in particular those of silicate and nitrate, and positively correlated with the ratio between inorganic N and P (N inorg/P inorg), temperature, and the logarithm of cell abundance. Concerning this last variable, it must be noted that, since cell abundance appears as a denominator in the calculation of toxin content per cell, the null hypothesis would be a negative rather than a null correlation; in our case, the relationship was positive. No link was observed between TOC (total organic carbon) and the cell toxin content.

Concerning the toxin analogues detected in the *O.* cf. *ovata* bloom (Figure 2), OVTX-a and -b were present in all the samples between July 20th and October 3rd with concentrations ranging from 8.7 to 34.0 and from 3.5 to 14.1 pg_tox_ cell^−1^, respectively (Figure 2a). OVTX-c, -d, and -e appeared between July 20th and September 13th (except on August 8th and 28th) with concentrations up to 3.1 pg_tox_ cell^−1^ (Figure 2b,c). OVTX-f and -g were detected only on July 24th and 27th, with concentrations between 0.21 and 0.38 pg_tox_ cell^−1^ (Figure 2c). Finally, isoPLTX was detected only between July 24th and August 1st, i.e., during the last period of the exponential phase of the bloom, with concentrations between 0.17 and 0.34 pg_tox_ cell^−1^ (Figure 2e). The toxin profile (Figure 3a, Appendix A) showed the dominance of OVTX-a as the most abundant toxin analogue, accounting for ca. 64–72% of the total content, followed by OVTX-b which represented 20–33% of total toxin abundance. OVTX-d was the third most abundant analogue, contributing less than 7%, followed by OVTX-c and -e, with a contribution of 1–6%. Note that these three analogues, OVTX-c, -d, and -e, were absent on three sampling days while OVTX-f, -g, and isoPLTX were detected in two or three samples and accounted for less than 2% of the toxin content.

### 2.2. Experimental Series 1: Comparing the Responses of the NW Mediterranean and the Adriatic Ostreopsis cf. ovata Strains to P and N Deficiency

Under f/2, control, and P-deficiency conditions, *O*. cf. *ovata* cultures (strain OOBM18) grew exponentially until day 7, while in the N-deficiency treatment, the exponential phase lasted only until day 4 (Figure 4a). Later, all cultures entered the stationary phase. The exponential growth rate (d^−1^) and the maximum cell yield (cells mL^−1^) were different in each treatment (Table 1). The highest growth rate was estimated in the f/2 treatment (0.39 d^−1^), followed by the control and the P-deficiency ones (0.32 d^−1^), with the lowest rates measured in the N-deficiency medium (0.17 d^−1^). The maximum cell yield was obtained in the f/2 medium, while the lowest was found in the N-deficiency treatment. Compared to the Adriatic strain (Figure 4b, Table 1), the NW Mediterranean *O*. cf. *ovata* strain had a lower growth rate and cell yield in the control and the N-deficiency media and similar values in the P-deficiency one.

As revealed by plotting the cell numbers that described the growth curves (Figure 4a), samplings for toxin and biovolume determination were conducted during different culture phases, i.e., exponential, intermediate, and stationary, which varied among treatments. This resulted in a different number of samples obtained per growth phase in the different treatments (Appendix A). Note that the comparison among treatments is made considering the physiological growth phase and not the sampling day.

In general, *O*. cf. *ovata* cells had the typical drop shape of the species during the exponential growth phase, when cells divide fast, and became round and aberrant (cyst-like forms) in the stationary phase, especially under P or N deficiency (not shown). In all the treatments, cell size parameters (dorsoventral (DV) and cell width (W)) showed a wide range of values, which always exhibited a unimodal distribution (DV median data in Appendix A). Within each treatment, DV and biovolume (Table 1) did not show marked differences between the different culture phases. Among treatments, there were also no noticeable differences in DV and biovolume, except for the f/2 culture, with lower cell size and biovolume parameters than the other culture media.

Total cell toxin content increased from the exponential to the stationary phase in all the treatments (Figure 5a). Overall, on a per-cell basis, under the f/2 high-nutrient condition, cells showed less toxin content (104 ± 43 pg_tox_ cell^−1^) than in the P-deficiency or N-deficiency treatments (Table 1). The control treatment gave the lowest average toxin content per cell in the exponential phase but presented similar values to the others in the stationary phase (Figure 5). An analysis of variance of toxin content per cell (after log transformation) with treatment (f/2, control, P-def, and N-def) and culture phase (exponential or stationary) as factors gave significant results for both (*p* < 0.0001 and *p* < 0.0005, respectively), with higher overall toxin content in the stationary than in the exponential phase. During the exponential phase, P-def and N-def presented significantly higher toxin content than the f/2 and control treatments.

Toxin analogues OVTX-a, -b, -c, -d, and -e were detected in most phases (Figure 5b–f) while OVTX-f and -g were only identified in the stationary period (Figure 5g–h) with cell toxin concentrations up to 1.0 and 2.1 pg_tox_ cell^−1^, respectively. IsoPLTX was detected in the stationary phase of all treatments and in the intermediate phase of the f/2 and control treatments, with estimated toxin concentrations up to 2.1 pg_tox_ cell^−1^ (Figure 5i). With respect to biovolume, the toxin concentrations were 6.4 ± 5.7 fg µm^–3^ in the f/2 medium, 4.3 ± 4.0 fg µm^–3^ in the control, 5.2 ± 5.3 fg µm^–3^ in the P-deficiency, and 5.4 ± 3.0 fg µm^–3^ in the N-deficiency treatments, respectively (not shown).

### 2.3. Experimental Series 2: Exploring the Effect of Different Inorganic and Organic N Sources at Two Different Concentration Levels on Ostreopsis cf. ovata Growth and Toxin Content

In the 50 μM N concentration series, the highest exponential growth rate (0.53 d^−1^) and cell yield (4549 ± 1437 cell mL^−1^) were measured in the cultures using urea as a N source (Table 1, Figure 6a). Slightly lower values were found in the cultures grown on ammonium (0.49 d^−1^; 3074 ± 591 cell mL^−1^), nitrate (0.44 d^−1^; 2950 ± 524 cell mL^−1^), and fertilizer as N sources (0.34 d^−1^; 1285 ± 196 cell mL^−1^). Due to the low cell densities attained in the 0.5 μM N treatment series, the experiments lasted 7 days only and showed a poorly defined growth curve (Figure 6b). In consequence, in these treatments, the terms “growth rate” and “cell yield” (Table 1) should be considered as an approximation. The highest growth rate and cell yield were observed in the cultures with fertilizer (0.15 d^−1^, exponential growth on days 0–4, 604 ± 122 cell mL^−1^) and ammonium (0.09 d^−1^, exponential growth on days 0–7, 460 ± 75 cell mL^−1^). In the nitrate and urea media, after an initial decrease in cell abundance, a small exponential growth phase occurred (0.15 d^−1^, days 2–4; 0.08 d^−1^, days 4–7, respectively), but cell yield was poor (less than 300 cell mL^−1^). Overall, the growth rates and the cell yields in the 0.5 µM series were 3- to 4- and 10-fold lower than in the 50 μM N series, respectively.

As in the first series of experiments (Section 2.2), the cells exhibited a high diversity of cell sizes and forms. In this second series, no relevant temporal differences in biovolume (Table 1) or DV values (Appendix A) were found within treatments. Overall, the cells in the 0.5 μM N series were smaller than in the 50 μM N series, and these, in turn, were smaller than in experimental series 1 (Section 2.2). Cultures grown on 50 μM urea had the smallest biovolume range and average values compared to the other 50 μM treatments.

In both the 50 μM and 0.5 μM series of experiments, total toxin content showed a similar temporal trend when expressed either per biovolume (not shown) or per cell (Figure 7 and Figure 8). In the 50 μM N series, the total toxin content (Figure 7a) was lower in the exponential than in the stationary phase with concentrations in the latter up to 32.2 ± 15.1 pg cell^−1^ in the ammonium treatment, followed by the nitrate (27.3 ± 4.9 pg cell^−1^), fertilizer (26.3 ± 8.9 pg cell^−1^), and urea (15.4 ± 6.8 pg cell^−1^) ones. OVTX-a to -e were detected in the three different phases and treatments with concentrations between 0.18 and 29.3 pg cell^−1^ (Figure 7b–f), while OVTX-f and -g were only detected in the stationary period (Figure 7g–h) with estimated cell toxin concentrations between 0.04 and 0.22 pg cell^−1^. IsoPLTX was detected in the stationary phase of all treatments and in the intermediate phase of the ammonium medium with the estimated cell toxin content up to 0.32 pg cell^−1^ (Figure 7i). In the 0.5 μM N series, the total toxin content (Figure 8a) showed minor variations throughout the sampled days, with a slight increasing trend in the nitrate and fertilizer treatments over time and with estimated cell toxin concentrations between 5.4 and 15.5 pg cell^−1^, 3- to 4-fold less than in the 50 μM series. OVTX-a and -b were detected with levels up to 8.1 pg cell^−1^ (Figure 8b–c), while OVTX-c, -d, and -e were detected with cell concentrations up to 1.5 pg cell^−1^ (Figure 8d–f). OVTX-f, -g, and isoPLTX were not detected in any of the 0.5 μM N series samples. An analysis of variance with high (50 µM) or low (0.5 µM) nitrogen levels, two nutrient sources (nitrate and fertilizer; urea and ammonium were excluded from this analysis because their samples were only from the exponential phase), and culture phases (exponential, intermediate, and stationary) as factors, gave significant results for nutrient level and phase (*p* < 0.0001) but not for nutrient source. A post hoc Tukey test corroborated the significantly higher toxin content per cell in the 50 µM group (*p* < 0.0001) and indicated that toxin content was highest in the stationary phase and lowest in the intermediate one (*p* < 0.05). An additional analysis of variance performed for the high-nutrient-level group, with four nutrient sources (nitrate, urea, ammonium, fertilizer) and culture phases (exponential, intermediate, or stationary), was significant for both factors (*p* < 0.0001), with the toxin content highest in the stationary phase and lowest in the intermediate one and urea leading to less toxin content per cell than the other nutrient sources (post hoc Tukey tests, *p* < 0.05).

The average total toxin concentrations expressed per biovolume in the 50 µM treatment series were 2.3 ± 2.2 fg µm^–3^ in the nitrate, 1.9 ± 1.5 fg µm^–3^ in the urea, 2.5 ± 3.3 fg µm^–3^ in the ammonium, and 2.5 ± 2.4 fg µm^–3^ in the fertilizer cultures. In the 0.5 μM treatment series, the results were 1.5 ± 0.7 fg µm^–3^, 1.7 ± 1.0 fg µm^–3^, 1.4 ± 1.0 fg µm^–3^, and 1.5 ± 0.9 fg µm^–3^, respectively, in the nitrate, urea, ammonium, and fertilizer media (data not shown).

Taken globally (Figure 3b, Appendix A), in the series 2 experiments, OVTX-a was the most abundant analogue, accounting for 30 to 64% of the total toxin content, followed by OVTX-b, which contributed ca. 16–36%. OVTX-d was the third most abundant analogue, accounting for ca. 7 to 16% of the total toxin content. OVTX-c and -e contributed with 3–12% in all cultures with lower values in the case of the OVTX-c in the 0.5 µM tests. OVTX-f and -g contributed to less than 0.8% in all samples but were not detected in the 0.5 µM N series.

## 3. Discussion

The general aim of the present study was to increase understanding on whether, and if so, how, nutrient conditions can modulate toxin production during natural *Ostreopsis* blooms. To do this, the proliferation of *O.* cf. *ovata* in a hot spot in the NW Mediterranean was monitored in 2018. Overall, as shown by Figure 1 and Figure 2 and by the relationships presented in Appendix A, increases in the toxin content per cell coincided with the consumption of the major inorganic nutrients and with increases in temperature and cell abundance. Interestingly, silicate was the nutrient exhibiting the strongest (negative) correlation with toxin content; however, this relationship should not be interpreted as an indication of direct effects but rather as a reflection of the general changes in the coastal ecosystem that accompany the development of the *Ostreopsis* bloom throughout the summer season. The positive correlation of toxin content per cell with cell abundance and the negative relationship with nutrient availability agree with the experimental findings (see below) of higher toxin content per cell in the stationary phase of the cultures.

We also found that toxin concentration was negatively correlated with the ratio N inorg/P inorg (Appendix A). Variations in the inorganic nutrient ratio have often been linked to changes in cell toxicity [29]. However, it is difficult to disentangle the influence of the various factors in the field.

Temperature has been shown to be a key factor in the proliferation of *O.* cf. *ovata* in temperate waters [28,46,47], but the effect of temperature on toxin production is not fully understood, due to the few available data and inconsistencies in the results obtained [48,49]. In our field study, the positive correlation of toxin content per cell with temperature could reflect the seasonal increase in temperature throughout the period of bloom development, potentially in combination with some direct effect on toxin production.

A strain isolated from the 2018 bloom was used in laboratory experiments under different nutrient-controlled conditions. The experiments included, on the one hand, a design similar to that described by Vanucci et al. (2012), using an *O.* cf. *ovata* strain isolated from the Adriatic Sea, in order to allow, as far as possible, a comparison between strains. On the other hand, the response of the NW Mediterranean *O.* cf. *ovata* strain was studied in detail in a series of experiments using inorganic and organic nutrient concentrations within the range found during the natural bloom. Although multiple combinations of nutrients occur in nature, only some possibilities were considered here. The discussion of the study will first begin by analyzing the experimental results. With the insight obtained, a final discussion focused on natural blooms will be presented.

### 3.1. Comparing the Responses of the NW Mediterranean and the Adriatic Strains to P and N Deficiency: Cell Growth, Cell Size, and Toxin Content in Experimental Series 1

As expected, the *Ostreopsis* cf. *ovata* strain OOBM18 isolated from the 2018 bloom in the NW Mediterranean exhibited optimal growth in the high-nutrient conditions of the f/2 medium, which presented the highest growth rate and maximum cell yield among all treatments (Table 1). However, although the f/2 medium had eight times more N and five times more P than the control medium, the cell yield in f/2 was only 2.4-fold higher than in the control. This fact suggests that the growth of this strain in f/2 may be limited by other elements and will not only depend on the absolute levels of nitrate and phosphate.

The OOBM18 strain, growing in experimental series 1 under the same conditions (control, P and N deficiency, Table 1) used by Vanucci et al. (2012) [37] for the Adriatic strain, exhibited similar growth patterns but differed on the biovolume and cell toxin content responses to nutrient availability.

The two strains showed the same trends of exponential growth rate and maximum cell yield related to the nutrient conditions. Overall, the results indicate that the *O.* cf. *ovata* cell growth was affected more severely by N than by P deficiency, in agreement also with Vidyarathna and Granéli (2010) and Accoroni et al. (2015) [29,50].

In contrast, the response of the two strains in terms of cell size was different. While the cell volume of the Adriatic strain increased throughout the growth curve under P deficiency (Vanucci et al. 2012) [37], no significant differences in biovolume were observed within the exponential, intermediate, and stationary phases of each particular treatment in experimental series 1 and 2 conducted here with the OOBM18 strain (Appendix A). In addition, the OOBM18 cell size distribution was always unimodal, and no size class distinction could be made, in contrast with the Adriatic strain in Vanucci et al. (2012) and other field and laboratory studies where distinct small and large cell populations, likely with different reproductive capacity, had been observed [37,51]. This finding, obtained in all treatments, including those in experimental series 1 (Table 1) using the culture media tested in Vanucci et al. (2012) [37], could indicate certain physiological differences between strains. In any case, the OOBM18 *O*. cf. *ovata* strain presented a great variety of cell sizes and shapes also documented in the strains isolated in 2009 from the same beach of Sant Andreu de Llavaneres, Lopud bay (Croatia) [51], and North Aegean coasts (Table 2) [52]. In addition, the values of the morphometric parameters of the OOBM18 *O*. cf. *ovata* under the different experimental conditions and growth culture phases (details of DV and W presented in Appendix A) fell within the range found in other studies (Table 2).

The total cellular concentration of toxins in the OOBM18 strain increased from the exponential to the stationary phase (expressed as toxins per cell or biovolume) in experimental series 1, as found in Vanucci et al. (2012) and other studies [41,42,43]. In general, the different analogues also increased from the exponential to stationary phase, although some of them (OVTX-e, -f, -g, and isoPLTX) were not detected at the beginning of the experiments when the cell abundance was low. Although toxins are generally considered to constitute a variety of secondary metabolites that tend to accumulate as growth slows or ceases [62], it cannot be concluded that OVTX-e, -f, -g, and isoPLTX were absent during the exponential phase. Their concentration could have been below detection level due to the low biomass of the cultures at the beginning of the experiment. More studies should be conducted to explore this question.

Unexpectedly, the f/2 treatment, with higher nutrient availability, showed lower total toxin content per cell than the control treatment in the stationary phase. In addition, the trends observed in the control vs. P- and N-deficiency treatments in series 1 were markedly different from the results obtained by Vanucci et al. (2012) [37]. While that study on an Adriatic strain reported a 40% and 53% decrease in cell toxin content under P and N deficiency, respectively, compared to the control treatment, we found a higher toxin content in the exponential (but not in the stationary) phase of the P- and N-deficiency treatments of the NW Mediterranean OOBM18 strain experiments. The present results agree with those reported for several harmful flagellates, which increased their cell toxin content mainly under P deficiency, e.g., *Prymnesium parvum* [63], *Chrysochromulina polylepis* [64], *Protoceratium reticulatum* [65], *Gambierdiscus toxicus* [66], or *Prorocentrum lima* [67,68]. Thus, from experimental series 1, it can be concluded that the different *O.* cf. *ovata* strains exhibit different responses to nutrient conditions, a finding that highlights the complexity of understanding the underlying processes and the difficulties involved in experimental approaches that attempt to mimic nature.

### 3.2. Use of Different Inorganic and Organic N Sources at Two Different Concentration Levels in Experimental Series 2

The 0.5 µM N concentration in these experiments, chosen to match the low N levels estimated in the natural seawater throughout the natural bloom, was too low to support sufficient culture growth and did not allow for clear conclusions. In the 50 µM N treatments, the experiments revealed a similar growth of *Ostreopsis* cf. *ovata* in urea, ammonium, and nitrate, while the amino acid-rich fertilizer resulted in lower growth. The toxin content was highest in the stationary phase, as found in the series 1 experiment, and was curiously lower in the intermediate than in the exponential one; the reason for this latter observation is unclear. Overall, the results of the 50 µM series agree with those of Jauzein et al. (2017), which showed that an *O.* cf. *ovata* strain from Villefranche-sur-Mer presented a strong affinity for ammonium [69]. Interestingly, the lowest toxin concentrations were detected in the urea medium. The experiments reported here also corroborate the ability of *O.* cf. *ovata* to use different sources of inorganic and organic N, which allows the proliferation of the microalga in environments affected by urban wastewater and leachates from agriculture. Our findings also suggest that cell growth does not necessarily work in parallel with toxin production under the different (and variable) nutritional conditions that *Ostreopsis* may face in nature.

### 3.3. Toxin Profiles and Nutrients

As expected, the toxin profiles of the OOBM18 strain and the field samples were similar, containing OVTX-a to -g and isoPLTX analogues, with OVTX-a as a major contributor to the total cell toxin content (>46% in the experiments and >60% in the field samples). These profiles correspond to Profile #1 in Tartaglione et al. (2017) [45], with slight variability. The maximum total toxin concentrations were around 147, 244, 190, and 237 pg_tox_ cell^−1^, in the f/2, control, P- and N-deficient treatments, respectively. These toxin contents on a per-cell basis were higher than the values reported for other Mediterranean *O.* cf. *ovata* strains (7.5 to 75 pg_tox_ cell^−1^) [4,58,70,71] and quite similar to those found for a French Mediterranean strain (up to 300 pg_tox_ cell^−1^) [18] and the NW Mediterranean strain studied in García-Altares et al. (2015) (up to 250 pg_tox_ cell^−1^) [72]. Furthermore, it should be noted that the NW *O.* cf. *ovata* strain is 5-10-fold more toxic than the Adriatic strain studied in Vanucci et al. (2012) [37], an observation that constitutes another difference between them in addition to those mentioned above (Section 3.1).

OVTX-f, -g, and isoPLTX were detected mainly in the stationary phases of the cultures, when cell abundance was high, but only accounted for less than 2% of the total toxin concentration. This suggests that their concentration may often be close to the detection limit and can only be estimated when enough biomass is collected (as already pointed out in Section 3.1). In all the 0.5 µM culture media, OVTX-f, -g, and isoPLTX were not detected, probably because *O.* cf. *ovata* growth and cell yield were too low under the imposed nutrient conditions. However, the hypothesis that their detection could have been possible if a higher number of cells had been collected needs to be tested. Since no differences in the toxin profile among different phases were found (OVTX-a being a major component), no hypothesis can be formulated about the potential biosynthesis order of the different toxin analogues based on the presented results. Future studies are needed to progress in this direction, as presented by Han et al. (2016) on saxitoxin congeners produced by *Alexandrium fundyense*.

In order to understand the potential role of nutrient availability, a principal component analysis (PCA) was performed by pooling all experimental data together (Figure 9); the field data were not included because an important number of analogues were absent. The first two components of the PCA, PC1, and PC2 explained most of the variance, 84.4% and 14.3%, respectively (Figure 9a). PC1 was positively correlated with all toxin analogues. PC2 was positively correlated with OVTX-a to -e (with the highest values for OVTX-a and -b), negatively correlated with OVTX-f and -g, and slightly negatively correlated with isoPLTX.

High positive scores of PC1 were found in the stationary phase of series 1 (Figure 9b), which presented the highest toxin content of the data set. In the 50 µM series 2 treatments (Figure 9c), the lower toxin-per-cell content compared to series 1 was reflected in negative PC1 scores; in the stationary phase, with comparatively higher cell content than in the exponential and the intermediate phases, the scores were less negative. Consistently with this trend, PC1 scores were negative in all the treatments and growth phases of the 0.5 µM series 2 treatments, given their estimated lower biomass and toxin content (Figure 9d). In general, within each series, the scores of PC2 were similar among treatments and growth phases (Figure 9e–g). This trend could be understood as showing a lack of influence of nutrient availability on the qualitative composition (profile) of the cell toxin content. The only exception was the markedly negative PC2 score of the stationary-phase samples of f/2 (Figure 9e); the reason for this finding is unclear but could be related to some imbalances in the composition of this high-nutrient medium.

Apparently, the results obtained in this study may seem contradictory to other publications. However, direct comparisons cannot be made, given the different approaches used in each case. The model of *Ostreopsis* toxin production elaborated by Pinna et al. (2015) [40] was based, among others, on the experiments by Vanucci et al. (2012), and the differences between the Adriatic strain and that used here have been discussed above. Finally, a comparison with Pezzolesi et al. (2016) [43] is unreliable due to the use of different experimental setups and designs. Probably, the modulation of toxin production is a complex process, not directly controlled by the essential nutrients N and P. It is also important to determine the role of the high-molecular-weight, C-rich toxins produced by *Ostreopsis* and how they relate to photosynthetic pathways and the storage of C products. We hypothesize that these toxins could be precursors of the mucopolysaccharides involved in cell attachment to substrates. Specific studies on this issue are required.

### 3.4. Risk of Exposure to Ostreopsis cf. ovata Toxin Analogues

The bloom of *Ostreopsis* cf. *ovata* in 2018 showed a temporal variability in the total toxin content and in the presence of the PLTX analogues (Figure 1, Figure 2 and Figure 4, and Appendix A). However, the positive correlation (r = 0.78) between the logarithms of the total toxin content expressed per macroalgal biomass (i.e., µg total toxin per gram of fresh weight, gFW^−1^) and the epiphytic *O.* cf. *ovata* cell abundance (cells gFW^−1^) indicates that the toxicity in the natural environment was highest when the abundance of the microalga was high. Based on the experiments performed with the isolated strain OOBM18, it could be hypothesized that, although the nutrient source or stoichiometry may not play a major role in determining the toxin production per cell in the blooming area, a major factor to be considered is the spatial dimension of the *Ostreopsis* bloom and the cell concentrations attained. This points to the need to explore strategies to minimize the occurrence of *Ostreopsis* cf. *ovata* proliferations.

One necessary next step, although difficult to address, is to investigate how the different toxin analogues can be transferred from herbivores to upper levels of the food webs (e.g., carnivorous fish) and how toxins are bioaccumulated and potentially transferred to humans.

## 4. Conclusions

The potential link between nutrient availability and toxin production by *Ostreopsis* cf. *ovata* is explored by combining the monitoring of a natural bloom of this toxic microalga in a NW Mediterranean coastal site in 2018 with experiments using a strain (OOBM18) from this bloom. Although nutrients fuel *Ostreopsis* biomass, the cellular profile of toxin analogues does not appear to be strongly linked with N and P availability. The temporal variability in the concentrations of the different toxin analogues observed in the field and the laboratory does not allow us to hypothesize a biosynthesis order. Probably, some analogues, including isoPLTX, could be present below detection limits. It is hypothesized that the high-molecular-weight, C-rich toxins produced by *Ostreopsis* are related to photosynthetic pathways and to the storage of C products and could be precursors of the mucopolysaccharides involved in cell attachment to substrates. Specific studies to investigate the biosynthetic pathways of PLTX analogues and their physiology role are required.

The potential risk of food-borne poisonings through a hypothetical food web transfer will depend on the dimension of the surface colonized by the benthic microalgae and the cell concentrations attained during the blooms. Compared with other *O.* cf. *ovata* strains, the OOBM18 NW Mediterranean strain has a relatively high toxin content, which qualitatively fits in Profile #1 described by Tartaglione et al. (2017) [45]. Still, some physiological differences among Adriatic and NW Mediterranean strains are expressed in their responses to similar experimental nutrient conditions.

Finally, the present study corroborates the capacity of *O.* cf. *ovata* to grow using organic N sources and suggests that this organism can be considered as an indicator of anthropogenically altered coastal ecosystems in particular in relationship to eutrophication.

## 5. Materials and Methods

### 5.1. Field Study

The *Ostreopsis* cf. *ovata* bloom in the 2018 summer was monitored on the rocky beach of Sant Andreu de Llavaneres, located on the Catalan coast of Spain (41°33.13′ N; 2°29.54′ E). Sampling for the characterization of the bloom was carried out following the regular monitoring procedures applied since 2007 in the area as described, for instance, in Vila et al. (2016) and Berdalet et al. (2022) [12,13].

Plankton samples were collected in 1 L plastic bottles. For the estimation of *O.* cf. *ovata* cell abundance, ca. 200 mL was immediately fixed with neutral iodine Lugol’s solution while the rest was processed in the laboratory for toxin determination. For the characterization of the *O.* cf. *ovata* epiphytic benthic population, 5 to 20 g of the dominant macroalgae (usually consisting of a mixture of *Ellisolandia elongata* (J.Ellis and Solander) K.R. Hind and G.W. Saunders, *Halopteris scoparia* (Linnaeus) Sauvageau, *Jania rubens* (Linnaeus) J.V.Lamouroux and *Padina pavonica* (Linnaeus) Thivy) was carefully collected and transferred into a 250 mL plastic bottle, to which ca. 180 mL of in situ GF/F glass fiber (0.7 µm nominal pore) filtered seawater was added. These samples, along with those for nutrient analysis (see below) were placed in a cooler box filled with in situ seawater to avoid warming during the 90 min transport to the laboratory where the subsequent processing was conducted.

In the laboratory, the macroalgal samples containing the *O.* cf. *ovata* epiphytic populations were vigorously shaken for one minute and subsequently sieved through a 200 µm mesh. The exact volume of the percolated seawater was measured with a cylinder because this parameter is required to estimate the epiphytic cell concentration. The macroalgae were slightly dried to remove excess water, and the fresh weight was determined. A percolated water subsample, ca. 60 mL, containing the epiphytic microalgae community was fixed with neutral iodine Lugol’s solution for cell number estimation, and the rest of the water was filtered through duplicate GF/F glass fiber filters (between 10 and 100 mL per sample) to estimate toxin concentration.

For estimation of nutrient concentrations, seawater samples were collected in situ in 30 mL polypropylene plastic tubes (for nitrate, nitrite, ammonium, phosphate, total N and P) and in polycarbonate bottles for TOC (total organic carbon) determinations. The samples were frozen upon arrival in the laboratory and stored at −20 °C until analysis. Determinations of nitrate, nitrite, ammonium, and phosphate were conducted with an AA3 HR autoanalyzer (Seal Analytical) following Grasshoff et al. (1983) [73]. Total N and P were estimated with an AA3 autoanalyzer after previous digestion, and TOC concentration was measured by high-temperature catalytic oxidation using a TOC-L (Shimadzu) following Salgado and Miller (1998) [74].

### 5.2. Laboratory Cultures and Experiments

**Strain isolation and stock culture.** Culture experiments were conducted with the *Ostreopsis* cf. *ovata* strain OOBM18 isolated by the capillary pipette method [75] from the benthic samples collected during the 2018 bloom. After an initial growth on microplates, cells were maintained in sterile 50 mL tissue culture flasks (polypropylene, vented, treated, from BD Corning) standing horizontally in a thermostatic room at 23 °C with ca. 100 µmol m^–2^ s^−1^ irradiance under 12 h:12 h Light:Dark cycle. The temperature was chosen as the optimal for *O.* cf. *ovata* isolated from our sampling area. Stock cultures were grown with micronutrients added at an f/2 minus Si (hereafter referred to as f/2) concentration [76]. Selenium, commonly used in dinoflagellate cultures, was not included in the medium because it did not improve the growth of this strain.

**Experimental series 1: comparing the responses of the NW Mediterranean and the Adriatic *Ostreopsis* cf. *ovata* strains to P and N deficiency.** In this first series of experiments, four treatments were established following Vanucci et al. (2012) [37] (Table 1, Appendix A): control, P- and N-deficiency treatments. Because the OOBM18 strain had been maintained in f/2 medium since its isolation, a fourth treatment in this medium was included to compare the responses with the other three media. The control medium contained, respectively, 8- and 5-times lower N and P concentrations than f/2, rendering a balanced 16:1 Redfield N:P ratio, slightly lower than the 24:1 N:P ratio in f/2. The concentrations of P and N were, respectively, 5 times lower in the P- and N-deficient treatments than in the control ones, to ensure P or N limitation, respectively, of the post-exponential phases of the culture growth. Vitamins and the rest of the microelements were added at concentrations proportional to the non-deficient N or P concentrations in each treatment, as in f/2. Cells were adapted to each culture medium treatment through four weekly sequential transfers (from a first culture in f/2) before the experiments were run. All transfers were conducted during the exponential phase of the growth curve, usually between days 4th and 7th, and the initial cell density in the transferred flasks was about 300–350 cells mL^−1^. The experimental vessels consisted of 250 mL tissue culture flasks containing 200 mL of each medium, placed horizontally in the same temperature and light-controlled incubator as the strain stock culture.

Five replicate culture flasks were grown in parallel for each nutrient treatment (Appendix A): three flasks were used for determining cell numbers and obtaining the growth curves (on days 2, 4, 7, 10, 14, 18, 22, 25, and 30) and two flasks for toxin concentration and biovolume determinations (on days 2, 7, and 18). On day 30, the three first flasks were used for estimation of all the parameters, since they had more biomass. For toxin estimations, between 15 and 45 mL of culture was filtered through duplicate GF/F glass fiber filters and stored frozen at −80 °C until analysis. For cell number and biovolume determinations, 10 mL culture aliquots were fixed on neutral iodine Lugol’s solution.

**Experimental series 2: exploring the use of different inorganic and organic N sources at two different concentration levels for *Ostreopsis* cf. *ovata* growth and toxin content.** This second series of experiments was designed taking as reference the inorganic and organic N concentration range found in the bloom area (see Results, Section 2.1). Nitrate and ammonium were chosen as inorganic N sources and urea and a commercial fertilizer as organic N sources (Table 1, Appendix A). The fertilizer (with a high amino acid concentration, according to the manufacturer) was analyzed in the laboratory before running the experiments; it contained ca. 15% nitrate, ca. 40% ammonium, and ca. 45% organic N with negligible P content. Two series of treatments, starting with 50 µM or 0.5 µM of each N source, were conducted, corresponding to the maximum and minimum concentrations estimated in the field study. The inorganic P concentration was fixed at 1/5th of the P content in f/2, i.e., 7.26 µM, in order to avoid any P deficiency and impose N as the main limiting element. All treatments were conducted with cells adapted to the culture media, following an approach similar to that in experimental series 1 for the P- and N-deficient treatments. The experimental flasks contained 200 mL of medium, and the initial cell densities were 300–250 cells mL^−1^. For each 50 µM N form condition (Appendix A), 15 initial replicate flasks were grown to allow five samplings in triplicate on days 2, 4, 7, 10, and 15, for determining cell growth, biovolume, and toxin concentration. In contrast, in the 0.5 µM N conditions, tests conducted previously to the experiment evidenced the extremely limited growth of *Ostrepsis* in these treatments, which did not provide enough biomass to inoculate 15 flasks as done in the 50 µM series. For this reason, for each 0.5 µM N form condition, only six replicate flasks were grown, with two replicates per sampling day of the experiments (days 2, 4, and 7) to determine the same parameters as in the first series of experiments. For toxin estimations, between 100 and 180 mL was filtered through duplicate GF/F glass fiber filters and stored frozen at −80 °C until analysis. For cell number and biovolume determinations, 10 mL culture aliquots was fixed on neutral iodine Lugol’s solution.

### 5.3. Chemicals

PLTX standard (from *Palythoa tuberculosa*) was purchased from Wako Chemicals GmbH (Germany). For quantitative analysis, concentrations of the target compounds (isoPLTX and OVTX -a to -g analogues) were determined as their respective PLTX equivalents. The stock standard solution (100 µg mL^−1^) was individually prepared by weight in MeOH:H_2_O (80:20, *v*/*v*) in a precision balance (Mettler Toledo AG425, Columbus, OH, US). For quantification, calibration solutions of PLTX were prepared from the stock standard solution at concentrations ranging from 0.005 to 5 µg mL^−1^ in MeOH:H_2_O (80:20, *v*/*v*). All standard solution aliquots were stored at −20 °C until use. LC–MS grade (>99%) water and organic solvents (acetonitrile and formic acid) were from Sigma-Aldrich (Steinheim, Germany). The mobile phases were filtered through 0.22 µm Nylon membrane filters (Whatman, Clifton, NJ, USA) before use. Nitrogen gas (99.95%, from Linde, Barcelona, Spain) was used as sheath gas and auxiliary gas in the ionization source. The 25 mm GF/F glass fiber filters were purchased from Whatman (Clifton, NJ, USA). The Lugol solution used to fix culture samples for *Ostreopsis* cell counts was prepared following Andersen and Throndsen (2004), with potassium iodide (KI) and iodine (I_2_) from Merck (Darmstadt, Germany) [77].

### 5.4. Ostreopsis cf. ovata Cell Counts, Growth Rate, and Biovolume Estimations

*Ostreopsis* cf. *ovata* cell counting was conducted in a Leica DMi1 inverted microscope following the Utermöhl method [78]. Different sedimentation chambers were used depending on the cell densities: 50 mL or 10 mL for the plankton samples (expressed as cells L^−1^) and 10 mL or 1 mL Sedgewick–Rafter for the benthic ones (expressed as cells per gram of fresh weight of macroalgae, cells gFW^−1^). In the experiments, cell numbers and biovolume were measured with 1 mL Sedgewick–Rafter counting chambers. The sampling and counting method in the study had an estimated global coefficient of variation (CV%) of 20%.

The specific growth rate (µ, d^−1^) was calculated from the least-squares regression line of the natural logarithm of cell numbers versus time during the exponential phase of the cultures, identified in the semi-log plot [79,80]. When the exponential phase was identified between only two days, the formula μ = (LnN_i_ − LnN_0_)/(t_i_ − t_0_) was applied, N_1_ and N_0_ being the cell concentrations at time i (t_i_) and 0 (t_0_), respectively.

Cell biovolume was measured in the experimental samples in order to have an additional parameter to compare the Adriatic and NW Mediterranean strains. Cell size parameters were measured from images captured with a camera connected to the Leica DMi1 inverted microscope and processed with the free computer program ImageJ [81]. For each treatment and selected sampling day (Appendix A), the dorsoventral diameter (DV) and width (*W*) of 100 cells were measured. The anteroposterior diameter (AP) was calculated using a fixed DV/AP factor based on the data presented by Guerrini et al. (2010), given that the DV and W of the cells in the present experiments fell in the range of that study [41]. The mean of the DV and AP of the Adriatic (57.2 μm and 24.5 μm, respectively) and Tyrrhenian (40.2 μm and 22.8 μm, respectively) strains rendered an average factor (48.7 μm/23.6 μm = 2.06 = f) which was applied to each cell in order to estimate the corresponding AP (AP = DV/f). Calculation of biovolume (µm^3^ cell^−1^) was made with the assumption of ellipsoid shape using the following equation [82]:V = (π/6)·DV·W·AP

### 5.5. Toxin Extraction and Analysis

Toxin extraction was conducted on 2 mL of MeOH:H_2_O (80:20, *v*/*v*) (previously cooled down to −20 °C) with two 5 min ultrasonic pulses in an ice-cooled bath and final vortex shaking. After centrifugation of the extract (4500 rpm, 4 °C, 5 min), the supernatant was filtered through 0.22 µm Nylon membranes and stored in amber glass vials at −80 °C until their analysis by ultra-high-performance liquid chromatography coupled to high-resolution mass spectrometry (UHPLC–HRMS) following essentially Medina-Pérez et al. 2021 [83]. Extracted ion chromatograms for PLTX and OVTXs were obtained by selecting the most intense peaks of the toxin ion cluster.

### 5.6. Statistical Analysis

Principal component analysis (PCA) [84] of the experimental data set was conducted to summarize the relationships among the variability of the different toxin analogue concentrations in the laboratory experiments. First, the concentration values of each toxin analogue (OVTX-a to -g and isoPLTX) estimated in the exponential, intermediate, and stationary phases of each treatment were averaged. Then, the concentration data (x_i_) were transformed using the expression y_i_ = log (x_i_ + 0.1), where 0.1 is the minimum non-zero concentration measured. The PCA analyses were based on the correlation matrix among these average log-transformed concentrations of the toxin analogue content per cell in each culture growth phase and treatment. PCA was not conducted on field samples due to the low number of toxin analogues detected (Appendix A). Total toxin content (+0.1, as for the PCA) and cell abundance values were also log-transformed before correlation and analysis of variance calculations. Data analysis and statistical methods were performed with Systat and with R (4.0.3) packages “ggplot”, “corrplot”, “pheatmap”, “ggpubr”, “tidyverse”.

## Figures and Tables

**Figure 1 toxins-15-00188-f001:**
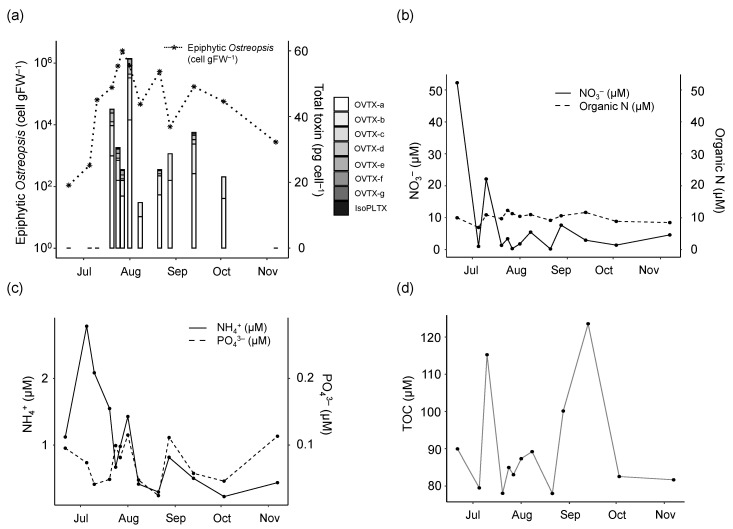
*Ostreopsis* cf. *ovata* 2018 bloom. (**a**) *Ostreopsis* epiphytic cell abundance per gram of macroalgal fresh weight (line, cell gFW^−1^) and cellular concentration of the different toxin analogues (OVTX-a, -b, -c, -d, -e, -f, -g, and isoPLTX) detected (bars, pg_tox_ cell^−1^). Small horizontal lines at the first three and the last sampling days indicate toxin levels below detection. (**b**–**d**) Main nutrient concentrations (expressed in µM): (**b**) Nitrate (NO_3_^−^) and organic Nitrogen; (**c**) Ammonia (NH_4_^+^) and Phosphate (PO_4_^3−^); (**d**) Total Organic Carbon (TOC).

**Figure 2 toxins-15-00188-f002:**
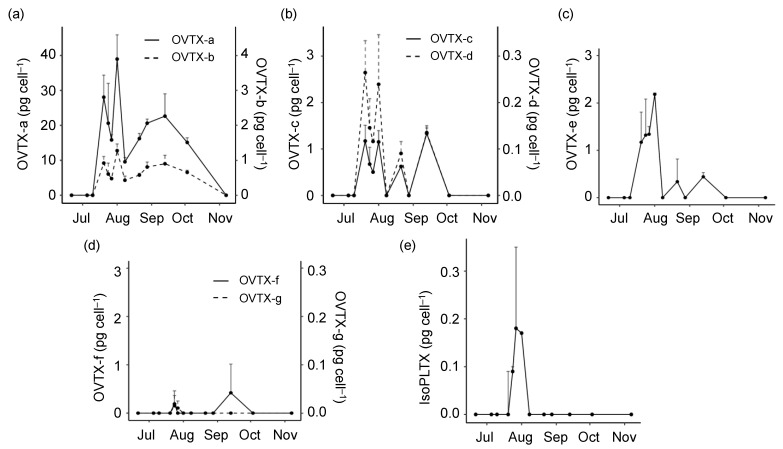
Cell concentration (average + standard deviation) of the different PLTX analogues (pg_tox_ cell^−1^) throughout the *Ostreopsis* 2018 bloom. (**a**) OVTX-a and OVTX-b, (**b**) OVTX-c and OVTX-d, (**c**) OVTX-e, (**d**) OVTX-f and OVTX-g, (**e**) isoPLTX.

**Figure 3 toxins-15-00188-f003:**
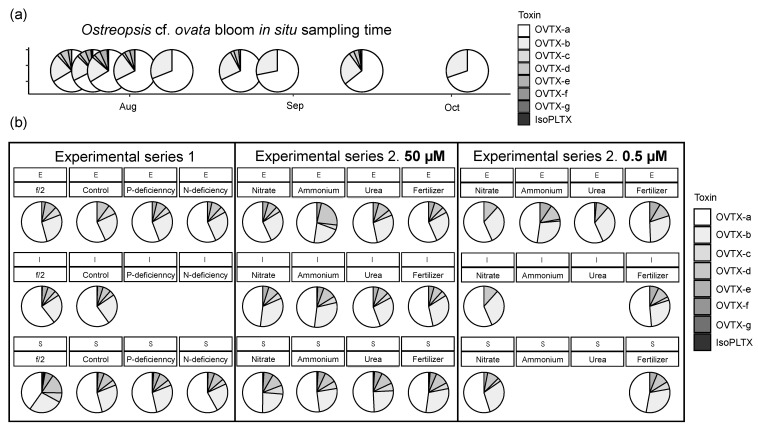
Pie charts of the toxin profiles of the samples collected (**a**) in the field throughout the *Ostreopsis* cf. *ovata* bloom in 2018 (each pie corresponds to the dot plot in Figure 1 and Figure 2) and (**b**) in the first and second series of experiments (E, I, and S indicate the Exponential, Intermediate, and Stationary phases of the growth phase of the cultures from which the samples were taken, as indicated in Appendix A). All profile data are presented in Appendix A.

**Figure 4 toxins-15-00188-f004:**
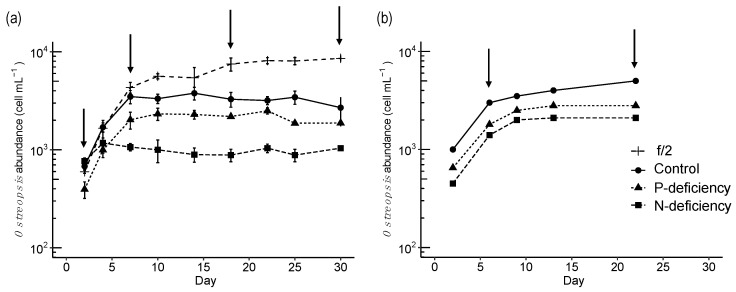
Experimental series 1. (**a**) Growth curves of the *Ostreopsis* cf. *ovata* strain from the NW Mediterranean under the different nutrient conditions in the first series of experiments of this study. Vertical bars indicate the standard deviation of the cell numbers, n = 2. Arrows point at the day when toxin content was estimated (see Figure 5). (**b**) Growth curves of the *O.* cf. *ovata* strain from the Adriatic Sea (redrawn from Figure 2 in [37]).

**Figure 5 toxins-15-00188-f005:**
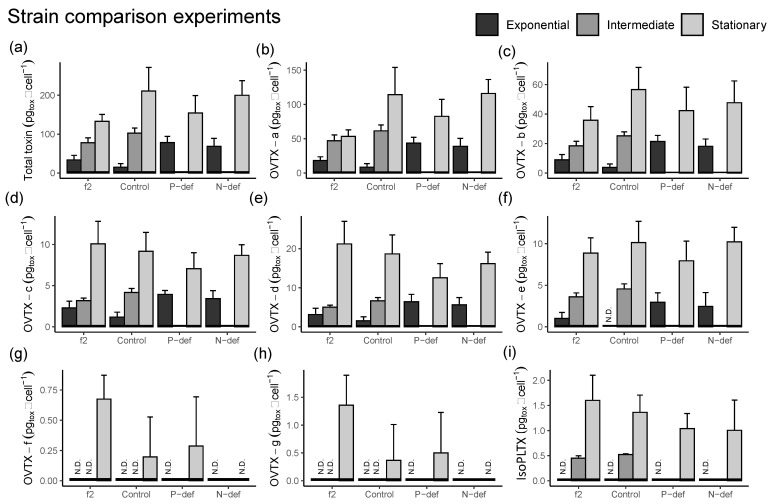
Experimental series 1. Toxin concentrations per cell (pg_tox_ cell^−1^.) of the *O.* cf. *ovata* cells grown in f/2, control, and P- and N-deficiency cultures in the present study. (**a**) Total toxin (sum of all analogues); (**b**–**h**) OVTX-a to -g. (**i**) IsoPLTX. The three culture phases (identified in Figure 4 and with samples listed in Appendix A) are represented by bars in each treatment: Exponential (black), Intermediate (gray), and Stationary (light gray). N.D: toxins not detected, i.e., below the detection limit. Vertical bars indicate the standard deviation; n varied as indicated in Appendix A. Note: There was no intermediate phase in either the P-def or N-def treatments.

**Figure 6 toxins-15-00188-f006:**
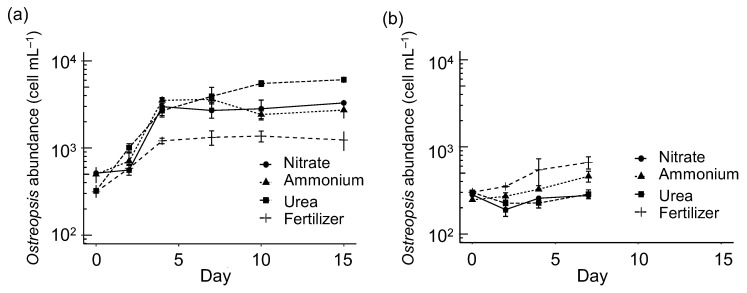
Experimental series 2. Growth curves of *Ostreopsis* cf. *ovata* in different N sources at (**a**) 50 μM and (**b**) 0.5 μM. Vertical bars indicate the standard deviation (*n* = 3 in (**a**), *n* = 2 in (**b**)). Note the shorter duration of the 0.5 µM series due to the scarcity of biomass.

**Figure 7 toxins-15-00188-f007:**
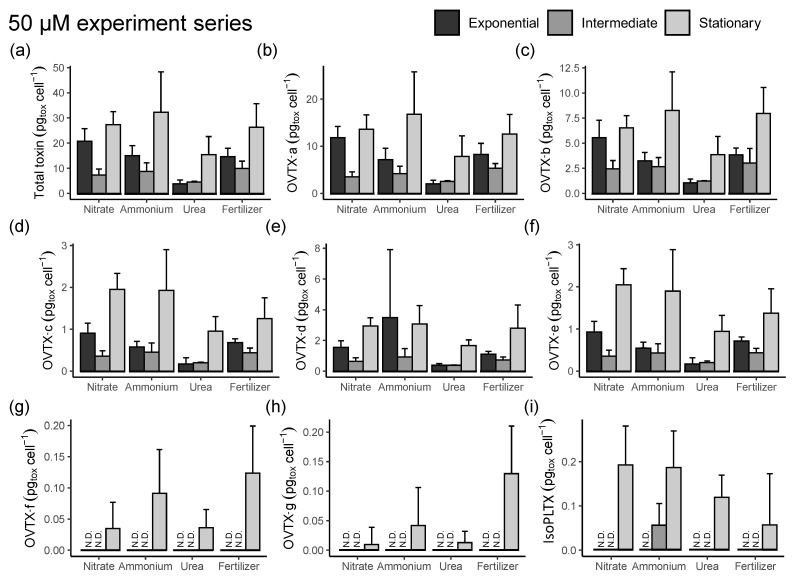
Experimental series 2. Toxin concentrations of *O.* cf. *ovata* cells grown on different N sources at 50 µM (pg_tox_ cell^−1^). (**a**) Total cellular toxin content per cell, (**b**) OVTX-a, (**c**) OVTX-b, (**d**) OVTX-c, (**e**) OVTX-d, (**f**) OVTX-e, (**g**) OVTX-f, (**h**) OVTX-g, (**i**) isoPLTX. N.D. indicates that the toxin levels were below detection. Vertical bars indicate the standard deviation; n varied as indicated in Appendix A.

**Figure 8 toxins-15-00188-f008:**
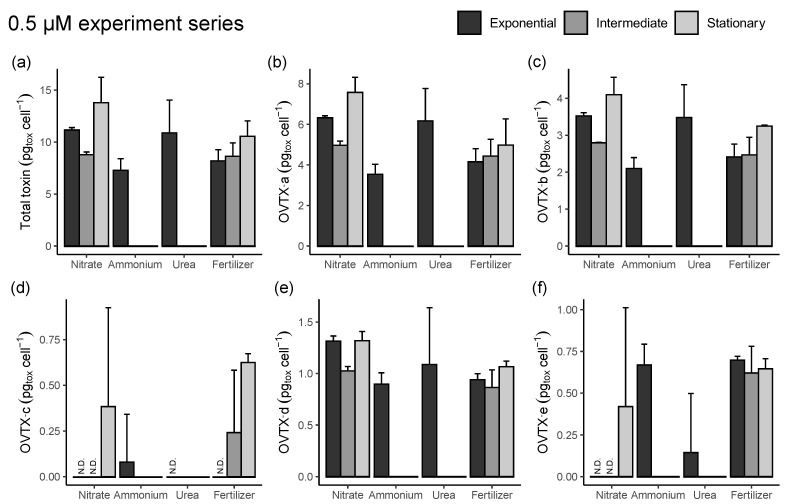
Experimental series 2. Toxin concentrations of *O.* cf. *ovata* cells grown on different N sources at 0.5 µM (pg_tox_ cell^−1^). (**a**) Total cellular toxin content per cell, (**b**) OVTX-a, (**c**) OVTX-b, (**d**) OVTX-c, (**e**) OVTX-d, (**f**) OVTX-e. N.D. indicates that the toxin levels were below detection. Vertical bars indicate the standard deviation; n varied as indicated in Appendix A. Note: In the ammonium and urea 0.5 µm treatments, the intermediate and stationary phases were not reached.

**Figure 9 toxins-15-00188-f009:**
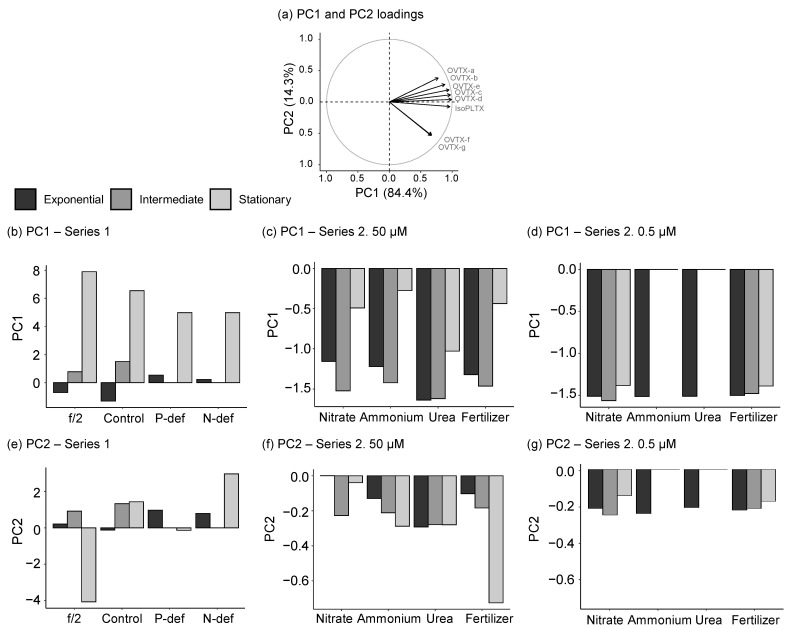
PCA analysis of toxin analogue concentrations in the different samples in culture experiments. (**a**) Loadings of the different toxin analogues on PC1 and PC2. The toxin contribution to each component is represented by the relative length of the arrows. (**b**–**g**) Scores of all samples (from the different treatments and growth phases) for PC1 (**b**–**d**) and PC2 (**e**–**g**). Note the different y-axis scales in (**b**–**g**). Notes: There were no intermediate phases in either the P-def or the N-def treatments. No intermediate or stationary phase was reached in the ammonium and urea (0.5 µm) treatments.

**Table 1 toxins-15-00188-t001:** Nutrient conditions in the different treatments, including the initial nitrate and phosphate concentrations of each culture medium and the corresponding N:P ratios. Summary of the growth characteristics of *O.* cf. *ovata* in each treatment, including growth rate (µ, d^−1^), maximum cell yield (average of the cell concentrations in all the samples obtained during the stationary growth phase of each culture), biovolume (µm^3^ cell^−1^, average ± standard deviation of the cell size measurements estimated during the experiments as described in Section 5.2 and Appendix A), and total toxin concentration per cell (average ± standard deviation of all samples analyzed per treatment).

Treatment	N Source (µM)	P Source (µM)	N:P	Growth Rate (µ, d^−1^)	Maximum Yield (cell mL^−1^)	Biovolume (µm^3^)	Cell Toxin Content (pg_tox_ cell^−1^)
Series 1. Comparing the responses of the NW Mediterranean and the Adriatic strains to P and N deficiency							
f/2	883	36.3	24	0.39	8058 ± 873	18,019 ± 8704	104 ± 43
Control	105	6.56	16	0.32	3299 ± 578	31,546 ± 9252	153 ± 91
*Control ([37])*	*105*	*6.56*	*16*	*0.39*	*5000*	30,000 ± 25,000	*25* ± 4
P deficiency	105	1.14	92	0.32	2118 ± 321	32,105 ± 13,013	140 ± 50
*P deficiency ([37])*	*105*	*1.14*	*92*	*0.30*	*2800*	30,000 ± 25,000	15 ± 3
N deficiency	30	6.56	5	0.17	935 ± 166	30,185 ± 9551	175 ± 62
*N deficiency ([37])*	*30*	*6.56*	*5*	*0.23*	*2100*	27,000 ± 15,000	*13* ± 2
Series 2. Exploring the use of different inorganic and organic N sources at two different concentration levels							
Nitrate 50 µM	50	7.26	6.87	0.44	2950 ± 524	14,718 ± 9705	22 ± 9
Ammonium 50 µM	50	7.26	6.87	0.49	3074 ± 591	14,861 ± 10,384	24 ± 15
Urea 50 µM	50	7.26	6.87	0.53	4549 ± 1437	8326 ± 8317	11 ± 8
Fertilizer 50 µM	50	7.26	6.87	0.34	1285 ± 196	11,888 ± 7053	21 ± 10
Nitrate 0.5 µM	0.5	7.26	0.07	0.15	267 ± 10	9640 ± 6577	11 ± 2
Ammonium 0.5 µM	0.5	7.26	0.07	0.09	460 ± 75	7522 ± 5188	7 ± 1
Urea 0.5 µM	0.5	7.26	0.07	0.08	287 ± 24	8850 ± 5603	11 ± 3
Fertilizer 0.5 µM	0.5	7.26	0.07	0.15	604 ± 122	9243 ± 7088	9 ± 1

**Table 2 toxins-15-00188-t002:** Morphometric characteristics of *Ostreopsis* cf. *ovata* taken from different references. Nd, no data; DV, dorsoventral diameter (length); W, width; AP, anteroposterior diameter (height); DV/AP, mean values.

DV (µm)	W (µm)	DV/AP	Sample Type	Reference
55–72	35–50	nd	Natural population from the French Atlantic Coast	[53]
14–62	14–44	nd	Cultures from natural population in NW Spain (Catalonia)	[54]
21–77	16–51	nd	Natural population from the NW Mediterranean Sea	[55]
19–75	13–60	2.31 ± 0.37	Natural population from the NE Adriatic Sea	[56]
24–85	17–65	nd	Cultures from natural population in NE Spain and Croatia	[51]
24–87	13–53	nd	Natural population from NE Spain and Croatia	[51]
29–68	17–46	nd	Cultures from natural population from the NW Adriatic Sea	[37]
40–65	18–45	nd	Natural population from Brazil (Rio de Janeiro)	[57]
48–65	31–46	nd	Natural population from the NW Adriatic Sea	[58]
28–76	18–51	nd	Cultures from natural population in the Adriatic Sea	[41]
22–59	14–44	nd	Cultures from natural population in the Tyrrhenian Sea	[41]
36–60	24–45	3.1	Natural population from the sea in Japan	[59]
30–71	18–53	2.4 ± 0.4	Natural population from the NW Adriatic Sea	[60]
26–62	13–48	14–36	Natural population from North Aegean Sea, Greece	[61]
19–60	17–52	nd	Cultures from natural population in the NW Mediterranean Sea: f/2 23 °C	Present experiment
33–69	24–55	nd	Cultures from natural population in the NW Mediterranean Sea: Control 23 °C	Present experiment
39–71	25–61	nd	Cultures from natural population in the NW Mediterranean Sea: P deficiency 23 °C	Present experiment
27–68	28–62	nd	Cultures from natural population in the NW Mediterranean Sea: N deficiency 23 °C	Present experiment
24–69	14–56	nd	Cultures from natural population in the NW Mediterranean Sea: Nitrate 50 µM	Present experiment
20–68	12–56	nd	Cultures from natural population in the NW Mediterranean Sea: Ammonium 50 µM	Present experiment
24–58	14–48	nd	Cultures from natural population in the NW Mediterranean Sea: Urea 50 µM	Present experiment
21–60	13–48	nd	Cultures from natural population in the NW Mediterranean Sea: Fertilizer 50 µM	Present experiment
28–60	15–52	nd	Cultures from natural population in the NW Mediterranean Sea: Nitrate 0.5 µM	Present experiment
17–57	11–43	nd	Cultures from natural population in the NW Mediterranean Sea: Ammonium 0.5 µM	Present experiment
21–55	15–47	nd	Cultures from natural population in the NW Mediterranean Sea: Urea 0.5 µM	Present experiment
19–64	11–57	nd	Cultures from natural population in the NW Mediterranean Sea: Fertilizer 0.5 µM	Present experiment

## Data Availability

Data are contained within the article.

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
