# Peer review of "Progress on the Link between Nutrient Availability and Toxin Production by Ostreopsis cf. ovata: Field and Laboratory Experiments"

_toxins, 2023, doi:10.3390/toxins15030188_

Round 1
Reviewer 1 Report
The authors studied the relationship between nutrient availability and toxin production of Ostreopsis cf. ovata using field and laboratory experiments. The study was well conducted and suggest that despite nutrients determine the intensity of the O. cf. ovata bloom, they would not modulate cellular toxin production. The study provided detailed comparison with literature studies that report different result on the relationship. The paper should include more mechanism studies/evidence (or at least discussion) to explain why nutrient does not modulate cellular toxin production and why their finding disagrees with previous reports.
Reviewer 2 Report
The study, with content suitable for the scope of the journal, investigated the amount of algae biomass and toxins generated by Ostreopsis ovata, and the toxin and profile production of algae isolated in the field under various nutritional conditions. The manuscript is somewhat lacking in completeness as a whole for the purpose of research, and appropriate revisions are absolutely required.
1. Please reset the thesis title appropriate for the research content.
2. Add the correlation between site water quality (N, P), algae abundance, and toxin content (profile) for the bloom period in 2018. CA, CCA, etc.
3. Explain the difference between the nutrient concentration in the field and the nutrient concentration used in the laboratory experiment. For this purpose, experiments with various nutrient concentrations (additional and limiting) to determine the amount of algae toxin and the profile are absolutely necessary.
4. Explain the changes in the toxin profile as well as the amount of toxin through the nutrient test. In general, the amount of toxin or profile (toxicity) varies depending on the type of nutrient and the amount added, and the order of biosynthesis is determined. See Marine Pollution Bulletin 104 (2016) 34–43.
Reviewer 3 Report
The work investigates the influence of the concentration and composition of nutrients on the bloom development of the benthic dinoflagellate Ostreopsis cf. ovata and on the concentration of the produced toxin in the cell. The research combines monitoring of the natural O. cf. ovata bloom and experiments using a strain from that bloom. The results are clearly presented and well explained. For the most part, previous findings on this subject have been confirmed, and the results are a valuable contribution to the general knowledge of phycotoxin production. Considering that this species often causes problems in the warmer seasons, and that beaches are also regularly monitored in some countries, I believe that the work will be of interest to the scientific community.
Round 2
Reviewer 1 Report
I am happy with the revision and suggest accept for publication.
Reviewer 2 Report
The authors responded relatively faithfully to the advice of the reviewers. Only a few minor things remain. Please explain this.
1. It is necessary to clearly differentiate between P-deficient and P-limited. Does P-deficient used in the study mean P-limited? Or did you simply mean a concentration lower than the in situ P? Or does it mean something else?
2. No correlation analysis was used to explain the relationship between nutrient availability and toxin production in this study. Express statistically significant or insignificant results using CCA, DCA, CA, etc. And we need to discuss why.
